# Microcycle Conidia Production in an Entomopathogenic Fungus *Beauveria bassiana*: The Role of Chitin Deacetylase in the Conidiation and the Contribution of Nanocoating in Conidial Stability

**DOI:** 10.3390/microorganisms13040900

**Published:** 2025-04-14

**Authors:** Rutuja Zambare, Vaidehi Bhagwat, Shivangni Singh, Maheswari Guntha, Vandana Ghormade, Santosh G. Tupe, Shamim Shaikh, Mukund V. Deshpande

**Affiliations:** 1R&D (Department of Scientific and Industrial Research recognized), Greenvention Biotech Pvt. Ltd., Uruli Kanchan 412202, India; rutujazambare@gmail.com (R.Z.); greenventionbiotech@gmail.com (S.G.T.); 2Rajiv Gandhi Institute of IT and Biotechnology, Bharati Vidyapeeth (Deemed to Be University), Pune 411046, India; shamim.shaikh@bharatividyapeeth.edu; 3Nanobioscience Group, Agharkar Research Institute, Pune 411004, India; vaidehibhagwat@aripune.org (V.B.); shivangnisingh@aripune.org (S.S.); maheswariguntha@aripune.org (M.G.)

**Keywords:** *Beauveria bassiana*, chitin deacetylase, microcycle conidiation, CNP and ACNP nanocoating, surface characterization

## Abstract

In the field, substantial quantities of insect pathogenic fungal conidia (5 × 10^12^/ha) are usually applied for the control of pests. In this regard, attempts are being made to obtain higher yields of conidia to make the process viable. One of the approaches is to induce microcycle conidia (MC) production. In a solid-state fermentation on rice, the SYB-grown inoculum with more pseudomycelia of *B. bassiana* enhanced MC production almost 5 times compared to the aerial conidia (AC) within 10 days. A chitosan (CNP) and alginate–chitosan (ACNP) nanocoating of MC increased the overall temperature and UV stability. The % cumulative mortalities of *Spodoptera litura* larvae over 10 d were 83 ± 8.0, 90 ± 5.0, 83 ± 5.0, and 90 ± 6 for AC-, MC-, CNP- coated MC and ACNP-coated MC, respectively. Using probit analysis, the LT_50_ values were 5.8, 6.0, 7.5, and 6.3 d for AC, MC, CNPs-MC, and ACNPs-MC, respectively. It was observed that chitin deacetylase (CDA) plays a significant role in increasing MC formation. The higher relative proportion of total CDA over chitosanase activity (higher CDA: chitosanase activity ratio) was found to be correlated with the microcycle conidiation.

## 1. Introduction

In fungus–insect interactions, most pathogenic insect fungi infect the host with asexual conidia. The aerial conidia (AC) germinate on the insect cuticle and invade the host. In the field, primary infection occurs when the conidia fall on the insect body, while secondary infection is when the conidia on the leaf surface attach to the insect body during crawling. In nature, the conidium germinates into the mycelium, producing aerial conidiophores bearing new conidia. Under certain conditions, the conidium germinates into a secondary conidium [1,2,3]. This is a process of asexual conidia formation in which the normal life cycle of the fungus is bypassed. For instance, high or low temperatures, pH, the salt concentration, and the presence of certain nutrients can stimulate the conidia to bypass the stages of vegetative growth. The shortened cycle to produce conidia directly from germinated conidia without intermediate hyphal growth is termed microcycle conidiation [4]. 

More than 100 fungal species exhibit microcycle conidiation. At higher temperatures, usually, fungi such as *Aspergillus niger*, *Aspergillus flavus*, and *Penicillium cyclopium* produce conidia on conidiophores bearing vesicles and phialides. In the case of plant pathogenic fungi like *Fusarium solani* and *Cercospora zeae-maydis*, most conidiophores produce conidia without the specialized phialides. In the case of pathogenic insect fungi such as *Metarhizium anisopliae* and *Beauveria bassiana*, a dimorphic transition is reported in which conidia germinate, and the germ tube elongation is quickly arrested; as a result, they directly differentiate into conidia [5]. 

One of the biochemical events that is so far correlated with microcycle conidiation in *M*. *acridum* is arginine metabolism [6]. According to them, nitric oxide synthase activity and nitric oxide levels are important in microcycle conidia (MC). Zou and co-workers reported the involvement of an endo-chitinase in conidial germination, conidial yield, and fungal resistance to UV-B irradiation and heat shock along with microcycle conidiation in *M. acridum* [7]. Earlier, it was reported that chitin deacetylase(s) (CDA) was constitutively produced by *M. anisopliae*, which significantly contributes to the softening of insect cuticles for penetration, self-defense against insect chitinases, and virulence [8]. It was observed that the proportion of chitosan, a deacetylated form of chitin, was higher in the unicellular yeast form cells than in the true filamentous form of a dimorphic fungus, *Benjaminiella poitrasii* [9]. 

In the present manuscript, we depict the role of CDA in microcycle conidiation in *B. bassiana* and highlight the significance of microcycle conidiation in mass production, viability, and virulence. The effect of a nanocoating with either chitosan or chitosan–alginate nanoparticles in stabilizing MC is also reported.

## 2. Materials and Methods

### 2.1. Fungal Strain

*Beauveria bassiana* NFCCI 3319 (ARI, Pune, India) was subcultured and maintained on the potato dextrose agar (PDA, potato, 20%; dextrose, 2%; agar, 2%). The stock culture was kept at 8 °C until use. The subculturing was performed every 15 d. 

### 2.2. Culture Conditions

The *B. bassiana* strain was grown on two different broths as well as agar media, viz. yeast extract–peptone–glucose agar (YPG, yeast extract, 0.3%; peptone, 0.5%; glucose, 1.0%; agar, 2%; pH 6.0) and sucrose yeast extract agar (SYA, sucrose, 3%; yeast extract, 5%; NaNO_3_, 0.3%; KH_2_PO_4_, 0.1%; MgSO_4_, 0.05%; KCl, 0.05%; MnSO_4_, 0.001%; FeSO_4_, 0.001%; agar, 2%; pH 5.5) for AC and MC production, respectively. Using slide culture technique, the same media was used to monitor the growth under the microscope. The effect of metal ion addition, Co^++^ (CoCl_2_, 1 mM) and Zn^++^ (ZnSO_4_, 1 mM), as well as both combined (Co^++^ plus Zn^++^ 1 mM each), on chitin-degrading enzymes and microcycle conidiation was studied in both media. 

The YPG and SY liquid media were inoculated (200 mL in 1000 mL flask) and incubated under shaking conditions (150 rpm) at 28 °C for 48 h as an inoculum for solid-state fermentation (supplier for raw material was Chaitanya Agro Biotech Pvt. Ltd., Malkapur, Buldhana, Maharashtra, India). For enzyme activities, the biomass was harvested after 96 h, and the enzyme activities were estimated as mentioned below. The YPG and SYB-grown biomass were used as inoculums for the solid-state fermentation (SSF) on a specific variety of rice (Dash). The rice variety usually affects the reproducibility. Depending on the starch concentration, some varieties showed sticky clump formation during autoclaving. Therefore, we used ‘Dash’-raw rice (10.56 M starch content), while other varieties contained ~18.56 M starch content. We used one variety for reproducibility. Unicorn bags were used during SSF. The inoculum was prepared by inoculating aerial conidia (2 × 10^7^/mL) in YPG and SYB (200 mL) media and incubating under shaking conditions (150 rpm) at 28 °C for 48 h. The Unicorn bags (autoclavable, type/14 with single microvented filter of 0.2 mm, 2 kg capacity, and size of 64 × 36 cm (Unicorn Imp & Mfg. Corp., Plano, TX, USA), filled with 1.5 kg of water-soaked rice as a substrate, were autoclaved at 121 °C for 45 min, and after cooling, they were inoculated with 200 mL of vegetative biomass, mainly mycelial, and incubated at 28 °C and 70–80% RH up to 10 d. The conidia were harvested in 0.1% Tween 80 and filtered through a sieve to separate solid substrate particles and mycelial fragments. The conidial yield and viability were checked as described before [10].

### 2.3. Light and Fluorescence Microscopy

The conidial germination was seen under a light microscope. For light microscopy, conidial suspension (100 µL of 1 × 10^7^ conidia/mL) of *B. bassiana* was inoculated on YPG and SY agar, incubated at 28 °C, and observed at different time intervals. 

Fluorescence microscopy was performed according to a modified method proposed by Ayliffe and co-workers [11]. The true mycelium, pseudomycelium, AC, and MC were stained with 10 µL of fluorescein isothiocyanate–wheat germ agglutinin (FITC-WGA, 0.05 mg/mL; 20 mM of Tris HCl buffer containing 1 mM of CaCl_2_, MgCl_2_, and MnCl_2_; pH 7) for 30 min in the dark and then washed with buffer to remove excess stain. The samples were seen under a fluorescence microscope at 465–495 nm excitation and 515–555 nm emission and imaged.

### 2.4. Appressorium Formation in Aerial and MC of B. bassiana

The AC and MC suspensions (100 µL of 1 × 10^7^ conidia/mL) prepared in 0.1% (*w*/*v*) Tween 80 were inoculated separately in 5 mL of YPG liquid medium and incubated at 28 °C. The germination was observed with a light microscope at two-hour intervals. Upon initiation of germination, the nutrient source was removed via centrifugation at 10,000× *g* (10 min), as described earlier by Nahar et al. [12]. The germinated conidia were washed twice with sterile distilled water, and the pellets were suspended in 100 µL of sterile distilled water. These suspensions were separately placed at the center of polypropylene Petri plates, then sealed with parafilm and incubated at 28 °C with 70–80% RH. The plates were periodically observed under the light microscope for appressoria development over 24 h.

### 2.5. Surface Characterization of Aerial and MC

#### 2.5.1. Conidial Settling Time

The AC and MC settling time (ST) were determined [13]. The conidial suspensions (1 × 10^7^ conidia/mL in 0.1% (*w*/*v*) Tween 80) were prepared and adjusted to an initial absorbance of 0.600 at 540 nm. The cuvettes were allowed to stand for 6 h, and the absorbances were recorded at 1 h intervals. The experiment was repeated twice using freshly prepared conidial suspensions, and the settling rates were expressed in percentages (ST_50_).

#### 2.5.2. Adhesion to Polypropylene

The adherence of hydrophobic conidia to polystyrene was studied [14]. The conidial suspension (5 × 10^4^/mL) in 0.1 M of potassium phosphate buffer (at pH 3.0, 5.0, and 7.0) was used for the assay. The suspension (100 µL) was poured into polypropylene Petri plates and left to settle for 24 h. The supernatant was then removed by the insertion in a beaker with 1.5 L of deionized water and agitated at 1000 rpm. The plates were observed under a microscope to quantify the adhesion of cells using microscopic images after 2 h.

#### 2.5.3. Microbial Adhesion to Hydrocarbons (MATHs) Assay

The conidial surface hydrophobicity was determined using the method proposed by Smith et al. [15]. Briefly, AC and MC of *B. bassiana* were washed in phosphate urea magnesium sulfate buffer (PUM, g/L: K_2_HPO_4_, 22.2; KH_2_PO_4_, 7.26; urea, 1.8; MgSO_4_·7H_2_O, 0.2; pH 7.1). The conidial suspensions were adjusted to obtain 0.4 OD (A_470_) and dispensed with (3 mL) of PUM buffer into acid-washed glass tubes. Xylene (300 μL) was added to each tube, and the tubes were vortexed three times (for 30 s each), allowed to stand at room temperature for 15 min, and the non-polar xylene phase was carefully removed. The tubes were cooled to 5 °C, and residual xylene, if any, was removed. The tubes were brought back to room temperature, and the absorbance of the resultant conidial suspensions was determined at 470 nm. The hydrophobic index (HI) was calculated using the following equation:(A_470_ control − A_470_, xylene treated)/(A_470_ control)

### 2.6. Enzyme Assays

The extracellular and intracellular enzyme activities, viz. chitinase, chitosanase, and CDA of supernatant and biomass grown in different media, were estimated as described earlier [10]. The biomass was separated via centrifugation and washed with cold distilled water. The 1 g of biomass (wet weight) in 10 mL of 0.05 M acetate buffer, pH 5.5, was homogenized using a glass homogenizer. The homogenate was used to estimate protein and enzyme activities.

As described earlier, the chitinase activities in the supernatant and culture homogenate were estimated using phosphoric acid-swollen chitin as a substrate [10]. The phosphoric acid swollen chitin was prepared following the method outlined by Vyas and Deshpande [16]. In the reaction mixture, 1 mL of 0.7% acid-swollen chitin; 1 mL of 0.05 M acetate buffer, pH 5; and 1 mL of appropriately diluted cell homogenate were added and incubated at 50 °C for 1 h. The *N*-acetylglucosamine (GlcNAc) produced was estimated colorimetrically at 585 nm with p-dimethyl amino benzaldehyde (DMAB) [17]. One international milli unit was defined as the activity that produced 1 nmol of GlcNAc per min. The same procedure was followed to estimate chitosanase activity using phosphoric acid-swollen chitosan as a substrate. One international milli unit was defined as the activity that produced 1 nmol of glucosamine (Glc) per min.

The chitin deacetylase (CDA) activity was measured as described earlier [8], using phosphoric acid swollen chitin as a substrate. The CDA activity was estimated with 100 µL of 0.05 M sodium tetraborate buffer, pH 8.5; 100 µL of 1 mg/mL phosphoric acid-swollen chitin, and 50 µL of enzyme incubated at 37 °C for 30 min [18]. The reaction was terminated by adding 250 µL of 5% (*w*/*v*) KHSO_4_. For color development, 250 µL of 5% (*w*/*v*) NaNO_2_ was added and allowed to stand for 15 min, and then 250 µL of 12.5% (*w*/*v*) ammonium sulfamate (N_2_H_6_SO_3_) was added. After 5 min, 250 µL of freshly prepared 0.5% (*w*/*v*) 3-methyl-2-benzothiazoline hydrazone (MBTH) was added, and the mixture was kept in a boiling water bath for 3 min. The tubes were cooled under water, and 250 µL of freshly prepared 0.5% (*w*/*v*) FeCl_3_ was added and read spectrophotometrically at 650 nm. One international milli unit of enzyme released 1 nmol of glucosamine from acid-swollen chitin per min. 

All the enzyme activities were estimated in triplicate twice.

The protein was estimated using Lowry’s method (1951).

### 2.7. Nanocoaoating of B. bassiana MC

#### 2.7.1. Synthesis of Chitosan and Alginate–Chitosan Nanoparticles

Chitosan nanoparticles (CNP) were synthesized via ionic gelation method with dropwise addition of 12.5 mL of sodium tripolyphosphate solution (1.6 mg/mL) to 100 mL of chitosan solution (0.5 mg/mL in 0.1M acetic acid) on a magnetic stirrer. Further, the solution was stirred for 2 h and sonicated for 10 min. 

Alginate–chitosan nanoparticles (ACNP) were synthesized via pre-gelation method [19]. First, 5.25 mL of 18 mM CaCl_2_ solution was added to 100 mL of 0.05% sodium alginate solution dropwise on a magnetic stirrer. Then, 5.25 mL of 0.05% chitosan solution was added in a dropwise manner with continuous stirring. After addition, the solution was stirred for 10 min and sonicated for 10 min. The particle sizes were measured using Malvern Zeta Sizer (Nano ZSP-90, UK) and NanoSight, while the zeta potential was determined by Malvern (Worcestershire, UK). The particles (1 mL) were dried in triplicate in hot air oven to obtain their weight.

#### 2.7.2. Nanocoating of MC

*B. bassiana* MC was nanocoated with fluorescent CNPs and ACNPs. Briefly, 50 µL of fluorescent brightener (Sigma (Setagaya, Japan), 0.05 mg/mL in 150 mM of NaCl and 20 mM of Tris HCl buffer, pH 7) was added to 25 mg of nanoparticles (CNPs and ACNPs) and incubated in dark with continuous mixing for 2 h. After incubation, the nanoparticle solution was centrifuged at 12,000× *g* for 15 min, and the pellet was washed twice with 50 mM of acetate buffer (pH 5.5). The conidial suspension (1 mL, 1 × 10^6^ conidia/mL) was placed into 2 mL Eppendorf and centrifuged at 12,000× *g* for 10 min. After centrifugation, the supernatant was removed, followed by adding fluorescent nanoparticles (100 µL) and incubating in dark for 15 min. The conidia were washed twice with buffer via centrifugation to remove excess stain. Conidia were imaged under a fluorescence microscope at 340–380 nm excitation and 435–485 nm emission.

#### 2.7.3. Viability of AC and MC 

*B. bassiana* conidial viability was studied by measuring conidial germination on YPG agar. The conidial suspensions (0.1 mL, 1 × 10^3^ conidia/mL) were inoculated on YPG agar and incubated at 28 °C for 24h.

The effect of temperature (50 °C, 30 min, and 65 °C, 30 min) on the conidial germination and further mycelial growth (CFU) was studied using YPG agar inoculated with 0.1 mL (1 × 10^3^/mL) conidia and incubated at 28 °C up to 72 h.

The effect of UV light (36 w germicidal lamp, 0.6 m, 10 min) on the conidial germination and further mycelial growth (CFU) was studied using YPG agar inoculated with 0.1 mL (1 × 10^3^/mL) conidia and incubated at 28 °C up to 12 d.

#### 2.7.4. Viability of Nanocoated MC

The viability of nanocoated *B. bassiana* MC was studied by measuring conidial germination in YPG and SY broth at 28 °C. The suspensions (1 mL, 1 × 10^6^ conidia/mL) of nanocoated MC were centrifuged at 12,000× *g* for 10 min, the supernatant was removed, and 1 mL of YPG/SY broth was added into each tube and incubated for 12 h.

The effect of temperature (50 °C, 15 min) on the conidial germination and further mycelial growth was studied using YPG broth inoculated with 1 × 10^6^ conidia and incubated at 28 °C up to 96 h.

The effect of UV light (36 w germicidal lamp, 0.6 m, 10 min) on the conidial germination and further mycelial growth was studied using YPG broth inoculated with 1 × 10^6^ conidia and incubated at 28 °C up to 12 d.

### 2.8. Bioassays with Spodoptera litura

The larvae of *Spodoptera litura* were reared in plastic containers duly washed with ethanol (90%), and a filter paper was kept at the bottom. They were maintained at 25–28 °C and 65–95% RH. Tender castor (*Ricinus communis*) leaves were sterilized with aqueous sodium hypochlorite solution (0.5% *v*/*v*), washed with sterile distilled water twice, and fed to the first and second larval instars. The larvae were provided with surface-sterilized mature leaves from the third instar. Second- or third-instar larvae of *S. litura* were used in bioassays of *Beauveria* AC and MC and the nanocoated MC. To determine median LT_50,_ the day-wise mortality data were used at the concentration of 1 × 10^7^ conidia/mL, as reported earlier [12]. The LT_50_ was calculated using probit analysis. The conidial suspensions (1 × 10^7^/mL) were prepared in 0.1% Tween 80. Larvae were dipped individually in 10 mL conidial suspensions for 5–10 s, and 30 larvae were transferred to separate plastic containers containing filter paper and castor leaves. The leaves were changed every alternate day, and the larvae were kept at 25–28 °C and 65–95% RH for 10 d. Dead larvae were kept in sterile Petri plates containing moist cotton swabs for 3–7 days at 25–28 °C at 70–80% RH to allow for mycelial growth and conidial development over the cadavers. The experiments were performed in duplicate. 

### 2.9. Statistical Analysis

The probit analysis was used to calculate the median lethal time LT_50_ (the time required for 50% of the *S. litura* larvae to be killed) of AC, MC, CNP-MC, and ACNP-MC (1 × 10^7^/mL conidia) preparations [12].

## 3. Results

### 3.1. AC and MC Production

The *B. bassiana* AC were inoculated on YPG and SY agar media with and without Co^++^ and Zn^++^ metal ions and incubated at 28 °C for 4 d. As shown in Figure 1, on YPG, conidia germinated into true mycelium (48 h), which developed conidiophore-producing conidia (96 h). In comparison, on SYA, the conidia germinated into pseudomycelium (hyphal bodies/blastospores) (48 h), which produced budding conidia, thus bypassing germ tube formation (96 h).

The SYA supported 3 times more conidia production (4.7 ± 0.5 × 10^9^/plate) than YPG (1.71 ± 0.3 × 10^9^/plate) agar at 28 °C for 6 d. The addition of Zn^++^ in YPG, pH 6.0, increased the conidial number by >20%, but the count was negatively affected in the presence of Co^++^ (Table 1). However, a similar effect was not seen when Zn^++^ was added to the SYA medium. The AC and MC showed >90% viability (germ tube forming conidia) under shaking conditions at 28 °C after 16 h. This indicated that microcycle conidiation did not affect the viability of conidia.

The pH of the growth medium affects the conidiation. For instance, when the pH of YPG agar was adjusted to 5.5, the conidiation was 25% higher than the YPG, pH 6.0 (2.15 ± 0.4 × 10^9^ as compared to 1.71 ± 0.3 × 10^9^/plate, respectively). In the case of SYA, pH 6.0 decreased the conidia production by almost 50% (4.7 ± 0.5 × 10^9^ decreased to 2.4 ± 0.3 × 10^9^/plate). The conidia were harvested as described in the Section 2.

Further, the conidia production was scaled up via solid-state fermentation on rice. It was seen that if the YPG-grown inoculum containing true mycelial biomass was used, the conidia production was 3.6 × 10^11^ ± 0.3/kg rice within 10 d at 28 °C. Meanwhile, with SY-grown biomass, mainly with blastospores and pseudomycelium as an inoculum, the conidial count was almost 5 times more, i.e., 1.48 ± 0.15 × 10^12^/kg rice within 10 d (Table 2). Furthermore, when YPG-grown inoculum was used, there was no significant conidia formation within 4 d. When SY-grown inoculum was used within 4 d, the count was greater, i.e., 3.1 × 10^12^ ± 0.5/kg rice (Table 2). The decrease in the count during further incubation can be attributed to the conidial germination and mycelium formation. This can be useful for the cost-effective mass production of *B. bassiana* conidia for field performance studies.

### 3.2. Visualization of Cell Wall Staining with Light and Fluorescence Microscopy

The AC germinated via germ tube formation, while the MC germinated via short germ tube and budding. Fluorescence microscopic images of FITC-WGA-stained true mycelium showed uniform staining, while the cell walls of the pseudomycelium showed patchy and lesser staining with FITC -WGA compared to the true mycelium, which signified the increased presence of deacetylated chitin (Figure 2).

Both AC and MC were stained with FITC-WGA (Figure 3). The AC were single or in clumps, while the MC were found in budding chains.

### 3.3. Development of Appressorium in Aerial and MC of B. bassiana

The appressorium is a mechanical peg that allows the germ tube to enter the insect cuticle. The CDEs facilitate the entry of the organism by softening the cuticle. Figure 4 depicts the appressorium formation from AC as well as from MC, suggesting that MC also have the potential to infect insect pests. 

### 3.4. Surface Characterization of AC and MC

The hydrophobicity, which affects the roughness of the cell surface, was found to influence the rate of sedimentation [20]. The higher the roughness, the faster the sedimentation. The comparison of AC and MC of *B. bassiana* showed that the settling time (ST_50_) in 0.1% Tween 80 for AC was 1.8 ± 0.2 h, while that for MC was 40% greater, i.e., 3.0 ± 0.2 h, suggesting that MC were less hydrophobic compared to AC (Table 3).

The adhesion to the polystyrene or polypropylene assay is based on the fact that hydrophobic cells adhere more to hydrophobic surfaces. The adherence of AC to polypropylene at pH 7.0 was highest (67.4 ± 9.0%), decreasing at pH 5.0 (33.6 ± 3.0%) and lowest at pH 3.0 (17.6 ± 2.0%). A similar trend was seen in the case of MC of *B. bassiana*, i.e., 44.0 ± 2.0%, 22.5 ± 4.0%, and 16 ± 2.0% at pH 7, 5, and 3, respectively. The adhesion of AC was relatively higher (11–35%) at pH 3.0, 5.0, and 7.0 than MC, indicating more hydrophobicity to the AC (Table 3).

The MATH assay is based on the affinity of hydrophobic conidia for non-polar solvents with van der Waals interactions. The conidia are partitioned between two immiscible solutions (e.g., water and xylene). The AC showed 0.794 ± 0.13 HI, while MC was 0.67 ± 0.10, indicating a reduced (26%) hydrophobic surface compared to MC (Table 3). 

The AC and MC were incubated in YPG at 50 °C for 30 min and then incubated at 28 °C for 24 h to check the viability. The CFU % was 75 ± 2.0% and 66 ± 3.5% for AC and MC, respectively. After incubation at 65 °C for 30 min, the CFU after 48 h was 52 ± 5.0% for AC, while colony formation was delayed in MC, which showed 51 ± 2.0% CFU after 72 h (Table 3). As mentioned in Table 3, UV stability was marginally higher for AC (50 ± 1.0% CFU) than MC (45.5 ± 2.0% CFU)

### 3.5. Biochemical Correlates of MC in B. bassiana

The extracellular and intracellular chitinase, chitin deacetylase, and chitosanase activities were estimated in YPG and SYB media with and without metal ions after 96 h. The constitutively produced intracellular chitosanase and CDA were 3 times more prominent in the SYB medium than in YPG-grown mycelia. This may be attributed to the presence of different metal ions in SYB. Co^++^ and Zn^++^ addition negatively affected the chitosanase activity in both media. On the other hand, the CDA activities increased with the addition of Co^++^ and Zn^++^ in both YPG and SYB media. There was no significant effect on intracellular chitinase activity, per se (Table 1). The extracellular activities were not significantly affected by the addition of metal ions. Table 1 highlights the correlation between intracellular enzyme activities in YPG and SYB and microcycle conidiation. It was seen that the CDA: chitosanase activity ratio significantly affected conidia formation. Usually, a higher proportion of CDA (ratio > 30) is necessary to obtain more conidia (Table 1). The drastic decrease in the ratio negatively affected conidia production.

### 3.6. Nanocoaoating of Conidia

#### 3.6.1. Synthesis and Characterization of Chitosan and Alginate–Chitosan Nanoparticles

The synthesis of chitosan nanoparticles (CNPs) was performed via the ionic gelation method, which resulted in positively charged (+39.4 mV) particles with a size of 128 nm. Alginate–chitosan nanoparticles (ACNPs) were synthesized via the pre-gelation method, displaying a negative charge (−23.9 mV charge) and size of 130 nm.

#### 3.6.2. Visualization of MC Nanocoating

The nanocoating of *B. bassiana* MC was carried out with fluorescent brightener-tagged nanoparticles for visualization. Nanoparticles successfully coated the conidia uniformly, as the images showed a uniform blue coating of fluorescent brightener-tagged nanoparticles on the conidia (Figure 5).

#### 3.6.3. Viability of MC With and Without Nanocoating

The MC in SY broth showed 80–90% germ tube formation and 10–20% budding up to 16 h. Pseudomycelium was seen after 24 h. Later, the budding MC were observed. Interestingly, the MC coated with either CNPs or ACNPs showed budding up to 12 h in SY broth (Figure 6). The budding was 30% greater in ACNP than CNP nanocoatings. The conidia were slowly released after 24 h, and pseudomycelium formation was delayed until 72 h. Later, the budding MC were observed.

#### 3.6.4. Temperature and UV Stability of Microcycle Conidia With and Without Nanocoating

The stability experiments were carried out in YPG broth to improve nutrient accessibility to the nanocoated conidia. The MC with and without nanocoating when incubated in YPG broth at 50 °C for 15 min and then incubated at 28 °C for 24 h in all the mycelial growth was observed over 96 h. The biomass (wet wt./10 mL broth) for microcycle, ACNP-coated, and CNP-coated MC were 1.3 g, 0.65 g, and 0.6 g, respectively.

The MC with and without nanocoating in YPG broth were exposed to UV light (36 W germicidal lamp, 0.6 m distance, 10 min) and further incubated at 28 °C up to 12 d. In the case of ACNP-coated MC, mycelial growth was seen after 8 d. Less significant mycelial growth was observed for CNP-coated MC and MC without coating over 8 d. This can be attributed to the reduced porosity of the CNP nanocoating. The ACNP coating might be more protective for the MC. The biomasses (wet wt./10 mL broth) for microcycle, ACNP-coated, and CNP-coated MC were 0.7 g, 0.35 g, and negligible growth, respectively.

### 3.7. Spodoptera Litura Mortality with AC, MC and Nanocoated MC

The % mortality of *S. litura* second- and third-instar larvae was evaluated in the bioassays with *B. bassiana* AC, MC, and nanocoated MC. The % cumulative mortality was 83–90% in 10 d with all the treatments. The mortalities were 83 ± 8.0, 90 ± 5.0, 83 ± 5.0, and 90 ± 6% for AC, MC, and CNP-coated MC and ACNP-coated MC, respectively. Interestingly, the mortalities were between 40 and 50% after 5 d with AC and MC suspensions, while with nanocoated MC preparations, they were between 20 and 30%. As cumulative mortality with all the preparations was between 80 and 90% after 10 d, this can be attributed to the delayed germination of conidia when nanocoated. The LT_50_ values were 5.8, 6.0, 6.3, and 7.5 d for AC, MC, and MC coated with ACNPs and CNPs, respectively, calculated using probit analysis. 

## 4. Discussion

To control insect pests and fungal pathogens, aerial or submerged (blastoconidia) conidia are used as infective propagules of biocontrol fungi. Compared to blastoconidia, AC perform effectively in the field due to their inherent environmental stability [21]. The MC is the intermediate form as far as environmental stability is concerned. Zhang et al. (2010) reported a 4–5 greater MC yield in *M. acridum* with the SY medium [22]. In the present investigation, the (microcycle) conidial yield of *B. bassiana* NFCCI 3319 increased 3-fold more on the agar media. However, on rice, a 4-fold increase was observed within 10 d (Table 2). Another *B. bassiana* isolate showed a 3.5-fold increase in the conidial yield under identical experimental conditions. However, in the case of the present strain NFCCI 3319 in the solid-state fermentation on rice, the yield was enhanced by 20-fold over 4 d (Table 2). FITC-WGA is specific for chitin staining. The FITC-WGA staining of both the true mycelium and pseudomycelium showed distinct differences. The pseudomycelium showed patchy staining, indicating the random deacetylation of chitin (Figure 2). Similarly, the relative proportion of chitosan was seen to be higher in the conidia (Figure 3).

The adhesion of conidia to the cuticle is the first step in fungus–insect interaction. The hydrophobicity of *B. bassiana* AC, blastospores, and submerged conidia was studied [23]. It was reported that AC adhered rapidly to both hydrophobic and hydrophilic surfaces, while blastospores and submerged conidia bound poorly to hydrophobic surfaces [23]. In our studies, the *B. bassiana* AC showed more hydrophobicity measured as the settling time, adherence to polypropylene, and by the MATH assay than the MC (Table 3).

In fungus–insect interactions, a series of events occur, such as conidium germination, appressorium formation for penetration, and yeast-like cells for colonization in the hemocoel [12]. Among the diverse fungus–host interactions, the most crucial events are the attachment and penetration of the host [24]. Earlier, it was reported that appressorium formation in *M. ansiopliae* was one of the virulence factors that decreased due to repeated in vitro sub-culturing on artificial media [12]. Appressorium formation was restored due to insect passage. In the present studies, both AC and MC displayed appressorium formation, indicating that MC have the same mechanism for insect penetration (Figure 4).

In *Aspergillus*, conidiation is regulated by a central regulatory pathway containing three genes, *BrlA*, *AbaA*, and *WetA* [25]. Similar regulation was reported in *Penicillium* sp. and *Talaromyces* sp. However, most entomopathogenic fungi, such as *Metarhizium* sp. and *Beauveria* sp., do not use core regulatory pathways. Moreover, in *M. acridum*, three genes in the central regulatory pathway have not been detected. In *M. acridum*, deletion of *MaCreA*, a zinc factor transcription factor gene, resulted in delayed conidiation and a significant decrease in conidial yield [26]. The possible involvement of an endochitinase, *MaCts1*, in conidial germination, yield, stress tolerance, and microcycle conidiation in *M. acridum* was suggested [7]. The disruption of *MaCts1* affected the microcycle conidiation in the SY medium, which usually induces MC formation in *M. acridum*. 

Cell wall chitin degradation follows two pathways, either inducible chitinases (EC 3.2.1.14) or constitutively produced chitin deacetylase (EC 3.5.1.41) and chitosanase (EC 3.2.1.132). These enzymes are reported to be important in morphogenesis and differentiation [27]. These enzymes are either induced or repressed in the growth media with metal ions, such as Co^++^, Zn^++^, and Mn^++^, which also affect microcycle conidiation (Table 1). In the case of *M. anisopliae*, three extracellularly and constitutively produced chitin deacetylases were reported [8]. The possible roles in insect cuticle softening and self-defense by changing the cell wall from chitin to chitosan were suggested [8,28]. Three chitin deacetylases in *B. bassiana* were reported, which modulate conidial development and virulence [29]. Disruption of *CDA1* retarded vegetative growth, and *CDA2* conidial yield and viability were affected. In the present studies, a biochemical correlation between the chitin deacetylase activity of the pseudomycelium and microcycle conidiation in *B. bassiana* was seen. The increased chitosan contents in the pseudomycelium also supported the observation of the biochemical correlation between CDA activity and regulated non-polarized (budding) growth [30]. The higher relative proportion of total chitin deacetylase over chitosanase activity (higher CDA: chitosanase activity ratio, Table 1) was correlated with microcycle conidiation. However, further studies with purified chitin deacetylase(s) and/or knocking of the CDA gene(s) are necessary to establish a cause–effect relationship between the pseudomycelium and microcycle conidiation, per se. Indeed, the purification of CDAs and knockout studies in the future would support the current observations of microcycle conidiation. 

The temperature and UV stability of the MC at par with the AC is a critical consideration for the mass production and use of MC in the field. In the present case, the MC were encapsulated in the CNP and ACNP to increase their stability. The nanocoated MC were protected from the higher temperature and showed mycelial growth after 96 h.

The influence of different doses of UV light on the insect pathogenicity of *B. bassiana* and growth was studied [31]. The increasing doses of UV light decreased the infectivity of the fungus. Earlier, the effect of UV-B radiation on conidial germination of a number of *Beauveria* isolates from different hosts and climatic and geographic origins to identify a strain with high UV-B tolerance for effective pest control in the field was studied [32]. Furthermore, the effect of UV on the conidia and alginate-encapsulated conidia of *B. bassiana* and *M. anisopliae* was studied [33]. The conidia were reported to be protected when encapsulated. Similarly, in the present investigations, ACNPs-coated MC showed mycelial growth after 8 d on UV exposure compared to the uncoated conidia, which showed delayed mycelial growth (12 d).

In the field, large quantities of insect pathogenic fungal conidia (5 × 10^12^/ha) are usually applied for pest control to exert an effect comparable to chemical insecticides [34,35,36]. In this regard, attempts are being made to obtain higher yields of conidia to make the viable process for field studies. In the solid-state fermentation on rice, the SYB-grown inoculum of *B. bassiana* enhanced MC production almost 5 times compared to AC within 10 days. Interestingly, 10 times higher conidial counts were obtained within 4 days. From the effective process point of view, this is a significant achievement. The productivity per day for AC was 3.6 ± 0.3 × 10^10^, which increased to 7 ± 0.15 × 10^11^, i.e., 20-fold for MC. Furthermore, the stability of MC was comparable to that of AC. The use of nanocoatings increased the overall temperature and UV stability of MC. The conidial formulations using alginate and/or chitosan coating can be optimized for field performance studies. 

## Figures and Tables

**Figure 1 microorganisms-13-00900-f001:**
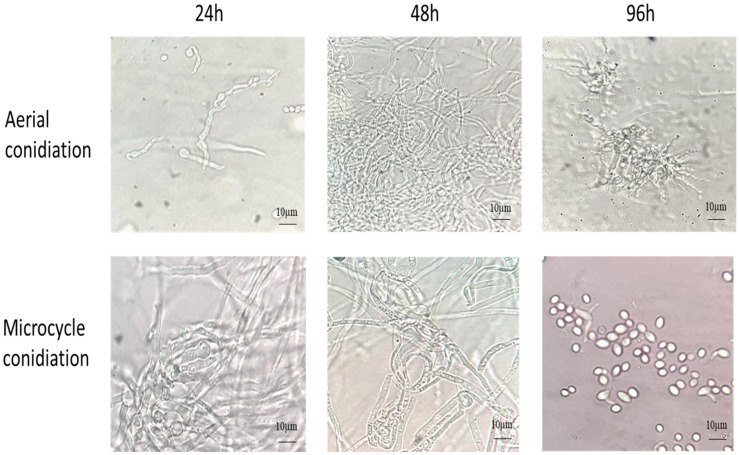
The germination of AC and MC of *B. bassiana* on YPG and SY agar, respectively, at 28 °C for 96 h.

**Figure 2 microorganisms-13-00900-f002:**
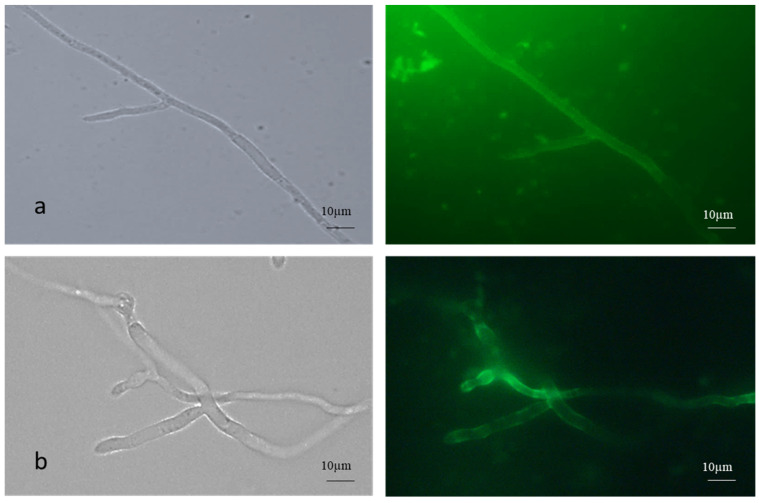
FITC-WGA staining of (**a**) true and (**b**) pseudomycelium of *B. bassiana* with brightfield and fluorescence microscopy images.

**Figure 3 microorganisms-13-00900-f003:**
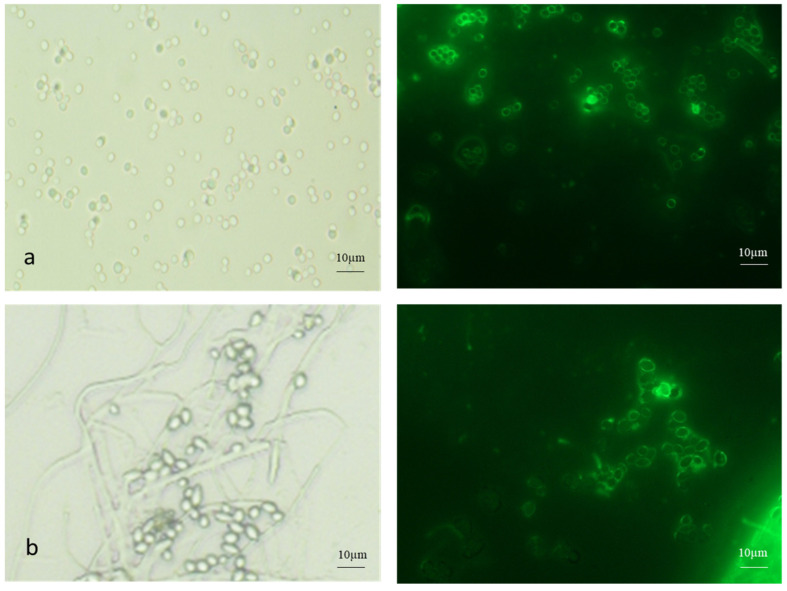
FITC-WGA staining of (**a**) aerial and (**b**) microcycle conidia of *B. bassiana* with brightfield and fluorescence microscopy images.

**Figure 4 microorganisms-13-00900-f004:**
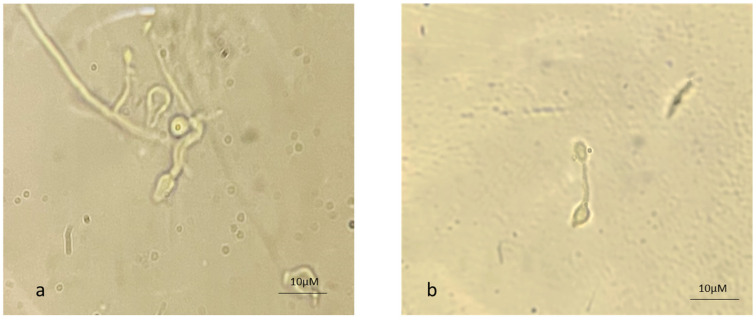
Appressorium formation from (**a**) aerial and (**b**) microcycle conidial germination in *B. bassiana*.

**Figure 5 microorganisms-13-00900-f005:**
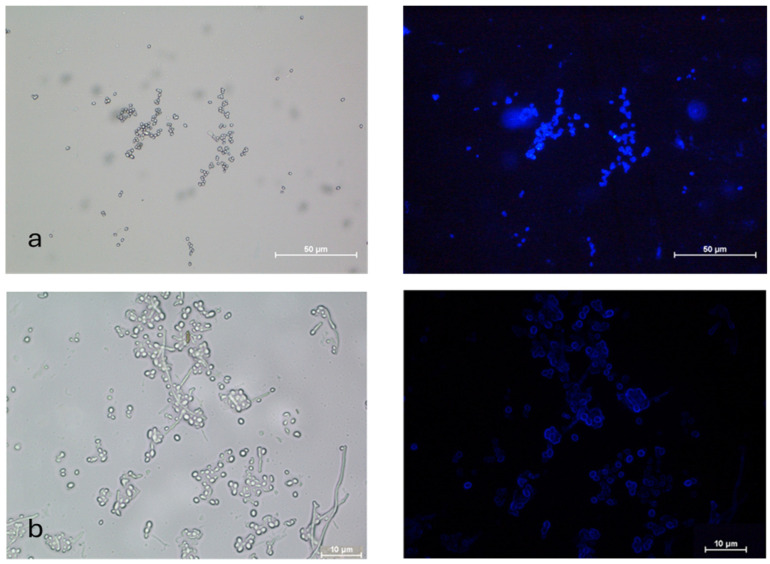
Microcycle conidia of *B. bassiana* coated with (**a**) fluorescent brightener-tagged CNPs and (**b**) fluorescent brightener-tagged ACNPs with brightfield and fluorescence microscopy images.

**Figure 6 microorganisms-13-00900-f006:**
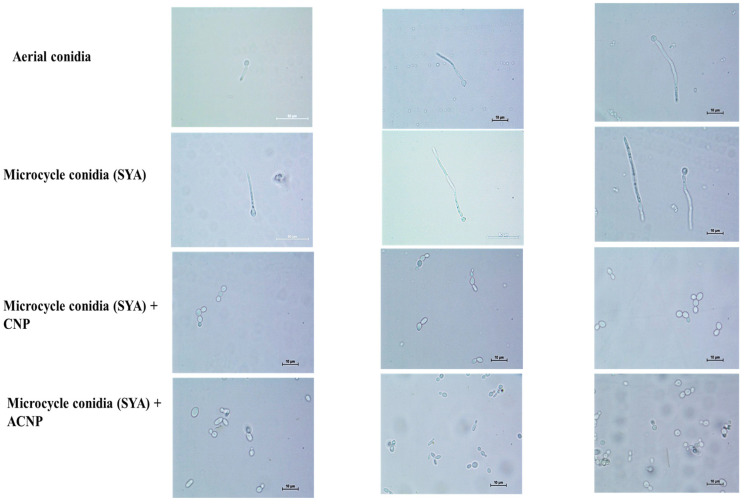
Nanocoated (CNP and ACNP) microcycle conidial germination up to 12 h.

**Table 1 microorganisms-13-00900-t001:** The intracellular chitinase, chitosanase, and chitin deacetylase activities of biomass of *B. bassiana* grown in different media after 4 d and conidial count after 6 d.

Medium	No. of	Chitinase	Chitosanase	CDA	CDA:
	conidia × 10^9^/				Chitosanase
	plate, 28 °C, 6 d				ratio
		mU/mg	mU/mg	mU/mg	
YPG, pH 6	1.71 ± 0.3	1.5 ± 0.2	6.41 ± 0.7	41.02 ± 3.0	6.39
YPG + Co^++^	0.4 ± 0.1	1.8 ± 0.2	159.00 ± 2.0	79.00 ± 32.1	2.38
YPG + Zn^++^	2.2 ± 0.4	1.96 ± 0.23	60.00 ± 1.2	405.00 ± 38.0	6.75
SYB, pH 5.5	4.7 ± 0.5	1.16 ± 0.4	26.45 ± 2.1	815.00 ± 14.0	30.81
SYB + Co^++^	0.8 ± 0.1	ND	100.00 ± 10.0	1530.00 ± 90.5	15.30
SYB + Zn^++^	4.5 ± 0.3	0.77 ± 0.3	40.00 ± 5.0	1240.00 ± 87.0	31.00

ND, not detected. The conidia were harvested as described in Section 2.

**Table 2 microorganisms-13-00900-t002:** Production of *B. bassiana* conidia in solid-state fermentation on rice.

Inoculum	Conidial Count/g	Conidial Count/kg	Conidial Count/kg
	Rice, 28 °C, 4 d	Rice, 28 °C, 4 d	Rice, 28 °C, 10 d
True mycelium	1.9 ± 0.2 × 10^8^	NS	3.6 ± 0.3 × 10^11^
Pseudomycelium	3.1 ± 0.3 × 10^8^	3.1 ± 0.5 × 10^12^	1.48 ± 0.15 × 10^12^

NS: not significant.

**Table 3 microorganisms-13-00900-t003:** The comparison of AC and MC of *B. bassiana* in terms of surface characteristics, temperature, and UV stability and virulence against *S. litura*.

Characteristics Settling Time (ST50), h	Aerial Conidia	Microcycle Conidia
Settling time (ST50), h	1.8 ± 0.2	3.0 ± 0.2
Adherence to polypropylene (%)		
pH 3.0	17.6 ± 2.0	16 ± 2.0
pH 5.0	33.6 ± 3.0	22.5 ± 4.0
pH 7.0	67.4 ± 9.0	44.0 ± 2.0
MATH assay (HI)	0.794 ± 0.13	0.67 ± 0.1
Temperature stability at 50 °C, 30 min, and incubation at 28 °C on YPG agar, 24 h	75 ± 2% CFU	66 ± 3.5% CFU
Temperature of 65 °C, 30 min, and incubation at 28 °C on YPG agar, 48 h	52 ± 5.0% CFU	ND (48 h),
	--	Delayed 51 ± 2.0% (72 h)
UV stability (36 W germicidal lamp, 0.6 m distance, 10 min)	50 ± 1.0% CFU	45.5 ± 2.0% CFU

CFU: colony-forming unit. ND: not detected.

## Data Availability

The original contributions presented in this study are included in the article. Further inquiries can be directed to the corresponding authors.

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
