# Peer review of "Microcycle Conidia Production in an Entomopathogenic Fungus Beauveria bassiana: The Role of Chitin Deacetylase in the Conidiation and the Contribution of Nanocoating in Conidial Stability"

_microorganisms, 2025, doi:10.3390/microorganisms13040900_

Round 1

Reviewer 1 Report (Previous Reviewer 2)

Comments and Suggestions for Authors

Some, but not all, of the problems in my previous review have been resolved. The authors provide a discourse on the relationship between chitin deacetylase (CDA) activity and microcycle conidiation (MC) in Beauveria bassiana and show evidence that high CDA:chitosanase ratio is correlated with high MC production. However, they do not provide a direct mechanistic connection between CDA and conidial stability, and additional studies with purified enzymes or knockouts might be needed. Statistical resolution is also an issue in that Table 3 does not have error bars or confidence intervals, and comparative strength for UV and temperature stability cannot thus be determined. The nanocoating chapter has encouraging laboratory results for heat and UV protection but no field validation, so one is left wondering if it will perform at all in the field. Figure 5 contains fluorescence microscopy images of nanoparticle-coated conidia, but no quantitative information on coating uniformity or coating adherence efficiency is given. Bioassays with Spodoptera litura rank MC as equally virulent as aerial conidia, with percentage mortality and LT50 values given but without statistical comparison between treatments formally tested. Discussion is still largely Metarhizium spp.-based, with little in the form of novel data added to that in the literature. Although the research delivers useful information regarding conidial yield optimization and stability, some important points—mechanistic validation, statistical stringency, and field relevance—have not been well developed.

Author Response

Reviewer 2 Report (New Reviewer)

Comments and Suggestions for Authors

Review Report

General Assessment

The manuscript addresses a relevant and timely topic in the field of biological pest control, focusing on enhancing the production and field stability of Beauveria bassiana microcycle conidia (MC). The study investigates the role of SYB-grown inoculum in enhancing MC production under solid-state fermentation on rice and evaluates the potential of chitosan (CNP) and alginate-chitosan (ACNP) nano-coatings to improve thermal and UV stability. Additionally, the research links enzymatic activity, specifically chitin deacetylase (CDA), to the regulation of microcycle conidiation. The study is innovative and has potential applications in improving the commercial viability of fungal biocontrol agents. However, the manuscript requires improvements in clarity, organization, and data interpretation to enhance its scientific impact.

Comments for Authors

Conidial Harvesting (Line 95): The method mentions harvesting with 0.1% Tween 80. Were conidia filtered or centrifuged after suspension? Were yields calculated per gram of substrate or per bag? These details are critical.

Media Composition (Lines 72-76): Recipes for YPG and SYA media are given, but the source of the ingredients (e.g., type and supplier of yeast extract, peptone, sucrose) is not specified. This affects reproducibility, as ingredient composition varies by supplier.

Bioassay Procedure Needs Clarification (Lines 223-238): The larval stage and size are indicated, but were larvae weighed or size-matched before bioassays? Only one repeat of the experiment is mentioned. Ideally, multiple independent replicates (biological replicates) are required.

How was mortality confirmed? Was mycosis checked for confirmation of fungal infection?

How were dead larvae handled to prevent secondary infection or cross-contamination?

There's no mention of statistical analysis methods in the Materials and Methods:

What statistical tests were used (ANOVA, t-tests, Kaplan-Meier for LT50)?

How many biological and technical replicates were included in each analysis?

Was any software used (e.g., GraphPad Prism, R, SPSS)?

Author Response

Please see the attached document

Reviewer 3 Report (New Reviewer)

Comments and Suggestions for Authors

Peer Review Report

 Title: The Microcycle Conidia Production in an Entomopathogenic Fungus Beauveria bassiana: The Role of Chitin Deacetylase in the Conidiation and the Contribution of Nanocoating in Conidial Stability."

Research Focus and Significance

The manuscript investigates Beauveria bassiana, emphasizing microcycle conidia (MC) production, the role of chitin deacetylase (CDA) in conidiation, and the impact of nanocoating on conidial stability. This research is highly relevant to biological pest control, as fungal conidia are critical to entomopathogenic effectiveness. By enhancing conidial yields, the study contributes to the scalability and commercialization of fungal-based biopesticides. Additionally, it offers valuable insights into the biochemical mechanisms governing conidiation and strategies to improve conidial stability under environmental stress.

Recommendation: The study tackles a significant challenge in biological pest control; however, the introduction should better articulate its advancement over existing research. A clearer connection to previous studies and a more explicit discussion of the study’s novelty are recommended.

Originality and Contribution

The study provides novel insights into the role of CDA in microcycle conidiation and the application of chitosan-alginate nanocoating to enhance conidial stability. The improved conidial yield achieved through a modified solid-state fermentation process represents a practical innovation. Additionally, the correlation between CDA activity and conidiation, along with the delayed germination observed in nanocoated conidia, significantly contributes to fungal biopesticide research.

Recommendation: While the study offers new perspectives, a more explicit comparison with existing literature would strengthen its impact. The novelty of the nanocoating approach should be evaluated against previously reported stabilization strategies.

Methodological Refinements

The study follows a well-structured experimental design, providing clear descriptions of fungal strain maintenance, culture conditions, enzyme assays, and Spodoptera litura bioassays. The nanocoating methodology is detailed, and microscopy techniques effectively support the findings. However, several aspects require further clarification. While the effects of pH, metal ions, and media on enzyme activity are well explored, the statistical analysis should be more rigorous, ensuring results are analyzed beyond standard deviation reporting. The UV and temperature stability experiments need additional details on exposure duration and intensity, with a direct comparison to similar studies to strengthen the discussion. In the bioassay section, it is essential to specify whether mortality data were analyzed using probit analysis or Kaplan-Meier survival analysis, as these are standard approaches in entomopathogenic fungal research.

Recommendation: A more robust statistical validation would enhance the conclusions. Additionally, the authors should justify the selected experimental conditions, particularly for the nanocoating process and bioassay setup.

Validity of Conclusions

The manuscript presents well-supported conclusions on the role of CDA in microcycle conidiation and the benefits of nanocoating in enhancing conidial stability. The correlation between enzyme activity and conidial yield is compelling, though further validation, such as gene knockout studies, would strengthen these findings. Some aspects require further refinement. While the delayed germination suggests that nanocoating improves conidial viability under stress, additional evidence from field studies or long-term viability assessments would be beneficial. Similarly, the role of CDA in microcycle conidiation is inferred from enzymatic activity measurements, but complementary molecular analyses, such as qPCR of CDA expression, would provide stronger support.

Recommendation: The conclusions should recognize the limitations of enzymatic correlations and the need for genetic validation. The discussion should also address potential trade-offs of nanocoating, including its impact on virulence and environmental persistence.

References and Data Presentation

The manuscript provides a comprehensive reference list covering key studies in fungal biology and biopesticide development, though some areas could be improved. Important studies on fungal nanocoating and conidial stabilization are missing, and recent advancements in nanoparticle-based fungal formulations should be incorporated. While the data presentation is generally clear, some tables, such as Table 1 and Table 3, contain extensive numerical data that would be more effectively visualized through graphs. Additionally, figures depicting conidial morphology and fluorescence staining are informative but should include clearer labels and scale bars for better interpretability.

Recommendation: The authors should improve data visualization by using graphs where appropriate and ensuring all figures are clearly labeled. Updating the literature review on nanocoating in fungal biopesticides would further strengthen the discussion.

Final Recommendation

The manuscript presents a well-executed study with significant implications for fungal biopesticide research. However, revisions are required to strengthen the manuscript’s clarity, rigor, and impact.

Recommendation: Major Revisions

Key areas for improvement include strengthening the discussion of novelty and how this research advances previous studies, enhancing statistical validation of experimental results, and providing greater clarity on methodology, particularly in UV/temperature stability tests and bioassay analysis. Conclusions should acknowledge limitations and propose further validation approaches, while data presentation should be improved by incorporating graphs where appropriate and ensuring clear figure labeling.

Comments on the Quality of English Language

The English could be improved to more clearly express the research.

Author Response

Not received

Round 2

Reviewer 1 Report (Previous Reviewer 2)

Comments and Suggestions for Authors

One of the basic weaknesses of the manuscript is the lack of direct mechanistic evidence for a connection between chitin deacetylase (CDA) activity and microcycle conidiation (MC) and conidial stability. While the authors suggest that a higher CDA:chitosanase ratio is correlated with increased MC formation, they do not provide any experimental data beyond references to the literature. Their conclusion, hence, is speculative and not evidence-based. In order to establish such a claim, additional biochemical evidence involving assays for enzyme activity across different growth conditions, comparative transcript profile analysis, or knockout/overexpression experiments for CDA will be needed. If options are not feasible, the conclusion must openly frame that they could only correlate to their data but never state causation outright. Without the explanation, the salient finding of this study stands unsupported.

The second point of major concern is the absence of statistical comparisons between the key data points, particularly the LT50 values for Spodoptera litura bioassays and conidia stability under UV and heat stress.
While the authors provide these values, they fail to perform statistical tests on these values to check if differences that are observed are significant or not. This is a flaw that undermines the validity of their results. To enhance this, the authors must conduct adequate statistical analyses, e.g., ANOVA of conidial stability values and log-rank tests of comparisons of LT50, ensuring levels of significance are adequately tabulated and presented in figures. If statistical comparisons cannot be performed, they should indicate this limitation clearly and justify why they were not performed. Without statistical validation, the data shown in the manuscript remain descriptive but not conclusive.

Finally, the paper lacks quantitative information regarding the efficacy of nanocoating on conidia.
While Figure 5 contains fluorescence microscopy photographs of coated conidia, they are qualitative photos and do not quantify the coating uniformity or adhesion. This raises a question about the uniformity and efficacy of the coating in real conditions. In order to exclude this issue, the authors need to submit quantitative data such as fluorescence intensity analysis for coating thickness measurement, SEM or AFM for visualizing coating uniformity, or comparing conidia mass and diameter before and after coating. In cases when it is not possible to perform such analysis, the authors need to state that and comment on its potential impact on the study conclusions. Without quantitative confirmation, the arguments concerning the efficiency of nanocoating are not whole.

These three—the lack of mechanistic verification for CDA, without statistics, and absence of quantitative nanocoating data—are the most serious limitations in the manuscript. These are most important to address to ensure the scientific validity of the research. If experimental verification cannot be performed directly, the authors should explicitly state their study limitations so that they are transparent and credible in their results.

Author Response

To enhance this, the authors must conduct adequate statistical analyses, e.g., ANOVA of conidial stability values and log-rank tests of comparisons of LT50, ensuring levels of significance are adequately tabulated and presented in figures. If statistical comparisons cannot be performed, they should indicate this limitation clearly and justify why they were not performed. Without statistical validation, the data shown in the manuscript remain descriptive but not conclusive.

These three—the lack of mechanistic verification for CDA, without statistics, and absence of quantitative nanocoating data—are the most serious limitations in the manuscript. These are most important to address to ensure the scientific validity of the research. If experimental verification cannot be performed directly, the authors should explicitly state their study limitations so that they are transparent and credible in their results.

Answer: Thank you very much for your candid opinion and valid comments to improve the  m/s. This study was initiated for the production of microcycle conidia in solid state fermentation and to explore the possibility to use  nano coating to stabilize conidia in the field.

  • In the present m/s the biochemical correlation of CDA with microcycle conidiation was shown. The supporting observation was the chitosan contents as visualized by FITC-WGA staining of true- and pseudo-mycelium. This observation could be useful in further studies to manipulate the nutritional conditions for higher CDA activity and to maintain proper ratio of CDA: chitosanase  

In the discussion we have mentioned (Line 464) “Indeed, the purification of CDAs and knockout studies in future would support the current observations of microcycle conidiation.”

  • At the time of field studies optimization of nano- coated conidia production, conidial stability under UV and heat stress and their performance in controlling pest will be addressed. 
  • We agree to the point that without statistical validation, the data shown in the manuscript remain descriptive but not conclusive. We have calculated the LT50 values using Probit analysis. In the future studies as mentioned above we will take care of all the necessary points regarding statistical analysis for LC50, conidial stability and other aspects.

Reviewer 2 Report (New Reviewer)

Comments and Suggestions for Authors

The authors have fully addressed all of my questions and comments. After reviewing the revisions and their responses, I have no further remarks.

I believe the manuscript is now suitable for publication in its current form. I congratulate the authors on their high-quality work and wish them a successful publication!

Author Response

The authors have fully addressed all of my questions and comments. After reviewing the revisions and their responses, I have no further remarks.

I believe the manuscript is now suitable for publication in its current form. I congratulate the authors on their high-quality work and wish them a successful publication!

Answer: We thank reviewer for the critical reading and the comments. Thanks for the encouraging and supportive comments

This manuscript is a resubmission of an earlier submission. The following is a list of the peer review reports and author responses from that submission.

Round 1

Reviewer 1 Report

Comments and Suggestions for Authors

The authors of the manuscript microorganisms-3373664 evaluate the role of chitin deacetylase in Beauveria bassiana microcycle conidiation and the effect of nanocoating on microcycle conidia. The manuscript has methodological flaws and inadequate research design.

Statistical analysis is missing — the authors claim that ”It was seen that % mortality was marginally lower  (66 ± 5 %) with aerial conidia than microcycle conidia (73 ± 7 %) after 48-72 h” (L357). The data were based on an experiment done with 15 individuals, with conidia suspension prepared with the same concentration of surfactant agent (0.1% Tween 80), although the author noted that ”lesser (40%) hydrophobic surface to microcylce (sic) conidia”. Therefore, it is possible to have different applied doses despite the same initial concentration. A correct experimental design must include more individual larvae (at least 30) and three experiment replicates. The data must be presented as a probit analysis (and not as a percentage).

The effect of nanocoating on microcycle conidia survival has not been evaluated using a correct research design. The authors used a qualitative test (budding in sucrose yeast extract ) and not the probit analysis.

The data are not designed and evaluated based on a statistical approach, specific to the biological systems.  

The authors assessed semi-quantitative (from + to ++++) the adhesion to polypropylene of aerial conidia. Hower, they claimed that ” The adhesion of aerial conidia was relatively higher (10-17%) at pH 3.0, 5.0 and 7.0 than microcylce (sic) conidia indicating more hydrophobicity to the aerial conidia”. How did they transform the difference from + and + to a percentage difference?

In Table 3, the authors claim to present the data for the MATH assay (HI). The values are discussed in L300-L301:” The aerial conidia showed 0.69± 0.13 HI while microcycle 300 conidia had 0.42 ± 0.1”, without being presented in Table 3.

The research design must be better tailored to the research scope and incorporate a statistical approach.

Reviewer 2 Report

Comments and Suggestions for Authors

Zambare et al. discuss the role of chitin deacetylase (CDA) in improving microcycle conidiation in the fungus Beauveria bassiana and its potential in biocontrol applications. The research is methodologically sound, using a three-pronged approach: enzymatic assays, nanocoating approaches, and insect bioassays for better conidial production, stability, and overall virulence. Most importantly, this has been tied together by rigorous experimentation linking CDA activity and microcycle conidiation. However, it lacks statistical clarity in some of the data presentations and mechanistic discussion of the role of CDA in conidial stability. The language is clear (8/10), with only very minor typographical errors. Figures are well designed; however, Figure 2 should be of higher resolution to show the differences in staining. The discussion is detailed but sometimes repetitive and would benefit from being more novel and insight-oriented. Overall rating: 82/100.

Majors:

While biochemical assays support the role of CDA in conidiation, direct mechanistic evidence is lacking. The study suggests that a high CDA:chitosanase ratio is necessary but does not establish a causal relationship (e.g., Table 1).

Statistical representations in Table 3 are inadequate for comparing UV and temperature stability; error bars or confidence intervals are missing, which weakens the reliability of conclusions.

While nanocoating use is novel, the efficacy of CNP and ACNP coatings in field conditions has not been explored. Results on UV stability (Fig. 6) are promising, but long-term biocontrol impacts remain untested.

Figure 5 shows nanoparticle coating but lacks quantitative data supporting uniformity or adherence efficiency. That would help the claims.

The bioassays using Spodoptera litura suggest that mortality is higher with microcycle conidia, but the statistical significance of the difference is not given, which precludes a broader interpretation of its practical effect.

Overemphasized in this discussion are the similarities of the role played by CDA in Metarhizium spp. rather than how this research brings an extension to existing information.

----------------

Minors:

The Materials and Methods section does not provide enough detail on enzyme activity normalization and possible confounders in solid-state fermentation, which may affect reproducibility.

SYB medium preparation description needs more detail in metal ion concentration and their interactions with the components of the medium.

Figure 4's legends should explain the importance of appressorium formation for pest control.

The manuscript should indicate whether increased conidial production using solid-state fermentation can be replicated for other strains of Beauveria.

There is a need to explain how hydrophobicity measurements, for example, the MATH assay, translate to field efficacy. The use of "nanocoating" in the manuscript could be better contextualized for industrial scalability.

-------------

The results of this study have strong implications for advancing fungal biopesticide technology, mainly in the optimization of production and stability. These findings on CDA and nanocoating open new promising avenues to enhance the commercial viability of B. bassiana in pest management. Further work is needed to validate these findings in a wide range of environmental conditions and with field trials. The work is important in terms of its contributions to microbial pest control, although it needs a bit more data to support the claims fully. Recommendation: Accept with moderate revisions, enhancing clarity, statistical rigor, and practical relevance.